# Concurrent Heavy Metal Exposures and Idiopathic Dilated Cardiomyopathy: A Case-Control Study from the Katanga Mining Area of the Democratic Republic of Congo

**DOI:** 10.3390/ijerph18094956

**Published:** 2021-05-06

**Authors:** Didier Malamba-Lez, Désire Tshala-Katumbay, Virginie Bito, Jean-Michel Rigo, Richie Kipenge Kyandabike, Eric Ngoy Yolola, Philippe Katchunga, Béatrice Koba-Bora, Dophra Ngoy-Nkulu

**Affiliations:** 1Unit of Training and Research in Cardiology, Department of Internal Medicine, University of Lubumbashi, Lubumbashi 07610, Democratic Republic of the Congo; kipengerichie@gmail.com (R.K.K.); yololaericy@gmail.com (E.N.Y.); ngoynkuludophra@yahoo.fr (D.N.-N.); 2Doctoral School for Medicine and Life Sciences, Hasselt University, 3590 Diepenbeek, Belgium; virginie.bito@uhasselt.be (V.B.); jeanmichel.rigo@uhasselt.be (J.-M.R.); 3Department of Neurology and School of Public Health, Oregon Health & Science University, Portland, OR 97007, USA; tshalad@ohsu.edu; 4Department of Internal Medicine, Official University of Bukavu, Bukavu 11102, Democratic Republic of the Congo; philkatch@yahoo.fr; 5Department of Internal Medicine, Service of Neurology, University of Lubumbashi, Lubumbashi 07610, Democratic Republic of the Congo; kobora2001@yahoo.fr

**Keywords:** environmental exposures, heavy metals, idiopathic dilated cardiomyopathy, Katanga Copperbelt

## Abstract

Blood and/or urine levels of 27 heavy metals were determined by ICPMS in 41 patients with dilated cardiomyopathy (DCM) and 29 presumably healthy subjects from the Katanga Copperbelt (KC), in the Democratic Republic of Congo (DRC). After adjusting for age, gender, education level, and renal function, DCM probability was almost maximal for blood concentrations above 0.75 and 150 µg/dL for arsenic and copper, respectively. Urinary concentrations above 1 for chromium, 20 for copper, 600 for zinc, 30 for selenium, 2 for cadmium, 0.2 for antimony, 0.5 for thallium, and 0.05 for uranium, all in μg/g of creatinine, were also associated with increased DCM probability. Concurrent and multiple exposures to heavy metals, well beyond permissible levels, are associated with increased probability for DCM. Study findings warrant screening for metal toxicity in case of DCM and prompt public health measures to reduce exposures in the KC, DRC.

## 1. Introduction

The Katanga Copperbelt (KC), with its significant copper and cobalt reserves, is a theater of intense mining and ore processing, producing more than half of DRC’s and about 6% of the world’s cobalt (the modern-day “oil” of a low-carbon economy) and copper, respectively [1,2,3]. Low-grade mining infrastructures compounding with artisanal mining, environmental degradation, and poor regulation account for concurrent and multiple environmental exposures to heavy metals and metalloids including but not limited to arsenic (As), cadmium (Cd), cobalt (Co), copper (Cu), lead (Pb), and zinc (Zn) [4,5]. Recent reports indicate that KC is one of the most polluted places in the world due to the contamination of water, soil, air, and food by toxic metals [6,7,8,9,10,11,12,13,14,15,16,17].

Previous studies in the KC have established an association between human exposure to toxic metals and disorders of neurodevelopment, decline in male and female fertility, and an increased occurrence of birth defects [18,19,20,21,22]. Other studies have shown that exposure to toxic metals confers a cardiovascular risk [23,24,25]. Toxic metals may have a direct effect on the functioning of cardiomyocyte membranes, ion channels, receptors, sarcomere, enzymes involved in the production of energy, and the antioxidant defense [26,27,28,29,30,31,32,33,34,35]. They may also lead to the dyshomeostasis of essential trace elements promoting adverse remodeling and ultimately dilated cardiomyopathy [36].

Recently in Lubumbashi, the largest mining city of the KC, severe cases of dilated cardiomyopathy burdened with high mortality were reported in relatively young patients, in contrast to findings in other populations. While risk factors such as hypertension, alcoholism, overweight and/or obesity, kidney dysfunction, atrial fibrillation, and difficult living conditions have been highlighted, the potential contribution of toxic metals to the high prevalence of idiopathic dilated cardiomyopathy (DCM) has been suspected previously [37]. Lubumbashi is growing in population with many challenges of urbanization and sanitation. There are many unpaved roads, dilapidated sewers dating from colonial times that are insufficient for draining of rainwater, and even effluents of mining companies that are located in the vicinity of homes. Garbage collection and recycling of dangerous products such as batteries and wrecks of vehicles is also a big challenge for the city. The majority of the population is poor, and artisanal mining is a major subsistence activity. Concurrent and multiple exposures to heavy metals, including some radioactive elements such as uranium, have been documented [14,38]. There is no universal health coverage, and the burden of major chronic diseases, including cardiac morbidities, lies on individuals [3,16,39,40]. In such contexts, identification of preventable causes of chronic illness and/or disability is of the utmost importance. In this study, we unveiled associations between idiopathic DCM and multiple exposures to heavy metals.

## 2. Subjects and Methods

### 2.1. Subjects

A total of 97 subjects seen prospectively for cardiovascular evaluations at the university clinic and the Lubumbashi’s Centre of Cardiology were consecutively enrolled in the study between November 2017 through January 2019. They were at least 16 years old, male or female, living in the general population of the KC. Of these 97, 68 (70.1%) had symptoms and signs of heart failure, elevated NT-proBNP, and DCM on echocardiography (either presumed idiopathic or postpartum cardiomyopathy). Twenty-nine subjects (29.9%) had no cardiovascular disease after cardiovascular check-up and were therefore kept as controls in the present study. They all have a normal 12-lead ECG. Of the 68 above subjects, those over 70 years, with known diabetes; renal failure with the need for dialysis; positive HIV test; and transthoracic echocardiography supporting rheumatic or degenerative valvular heart disease, pericarditis, cor pulmonale, congenital defects, suspected ischemic heart disease, or dilation attributed to hypertension, were excluded from the study (Figure 1).

### 2.2. Readout Parameters and Measurements

Demographic data, past medical history, signs, and symptoms of heart failure were collected using a standardized questionnaire.

Fasting routine blood (total blood cells count, INR, HIV test, sodium, potassium, magnesium, calcium, chloride, total cholesterol, HDL, LDL, triglycerides, urea, creatinine, uric acid, glycemia, AST, ALT, direct bilirubin, and indirect bilirubin) and urine tests (strip and microscopy) were performed in all subjects.

Transthoracic echocardiography was performed using a Vivid i ultraportable echo system (GE Medical Systems, Tirat Carmel, Israel). For each registration, five heartbeats were recorded. Cine loops were stored digitally and later analyzed in EchoPac version 113 software (GE Vingmed, Horten, Norway). Simpson’s biplane method was applied for the assessment of left ventricle volumes and ejection fraction. 2D parameters and conventional Doppler parameters were measured according to recommendations [41,42].

Electrocardiogram was obtained with Cardiax PC system (Imed, Budapest, Hungary) and interpreted according to The Minnesota code manual of electrocardiography [43] and a 24-h Holter-ECG was recorded by a DMS 300-4A Holter recorder (DM Software, Hunfelden-Dauborn, Germany). Data from each Holter assessment were processed using CardioScan software (Cardioscan GmbH, Hamburg, Germany).

A spot sample of urine, which was voided directly into 40 mL polystyrene vials with screw caps, was obtained from each patient. A blood sample was drawn by a trained nurse from a brachial vein into a 4-mL BD Vacutainer tube with spray-coated K2EDTA. Blood and urine samples were obtained the day of the first visit or the morning of the following day. Levels of 27 heavy metals were measured: lithium (Li), beryllium (Be), aluminum (Al), titanium (Ti), vanadium(V), chromium (Cr), manganese (Mn), cobalt (Co), nickel (Ni), copper (Co), zinc (Zn), arsenic (As), selenium (Se), molybdenum (Mo), palladium (Pd), cadmium (Cd), indium (In), tin (Sn), antimony (Sb), tellurium (Te), barium (Ba), platinum (Pt), mercury (Hg), thallium (Tl), lead (Pb), bismuth (Bi), and uranium (U). Palladium and mercury were only measured in the blood, while titanium and indium were only measured in the urine. Urinary concentrations of metals were reported in μg/g of urinary creatinine. A value of half the urinary limit of detection (LOD) of beryllium, vanadium, manganese, and antimony was attributed to respectively 2, 7, 9, and 1 participants whose concentration was below the LOD. All measurements were carried out at the Louvain Centre for Toxicology and Applied Pharmacology (Université Catholique de Louvain, Belgium), using Agilent 7500 ce instrument (for urine) and Agilent 7500 cx instrument (for blood). The same techniques already widely described were used [14,38,44]. Samples were anonymized, and analyses were performed blind.

### 2.3. Statistical Analysis

Shapiro–Wilk normality test was used to determine the distributions of metals levels, which were right-skewed, therefore summary results are presented as geometric means with their 95% CIs. The Zn/Cu ratio was calculated and included in the statistical analysis as previously of interest in other studies [45,46]. For the association studies, generalized linear models were used to obtain adjusted estimates, 95% confidence interval, and *p*-value. Graphs of the fitted models showing DCM probability were produced using genmod procedure with SAS 9.4 TS Level1 M5 2016 (SAS Institute Inc., Cary, NC, USA). The level of statistical significance was set at *p* < 0.05 (two-sided).

## 3. Results

### 3.1. General Characteristics of Study Participants

Appendix A gives the general characteristics of patients with DCM. On average, participants were 48 ± 14 years old. Three-quarters (78%) of them were hospitalized for an average of 14(8) days. As comorbidities and risk factors, 39% of them regularly consumed clay and 27% misused alcohol at the time of diagnosis. They were symptomatic with fatigue (93%), breathlessness (83%), orthopnea (83%), and early satiety (85%). Several of them were congested with pulmonary rales (49%), hepatomegaly (63%), sacral (66%), and legs edema (71%). They were functioning in NYHA classes III and IV in 71% with a severely reduced ejection fraction (21 ± 8 %). The initial treatment was essentially made of furosemide (95%) and ACE-i (85%).

Appendix A compares patients with controls according to demography, anthropometry, education level, and routine biology. On average, patients were less educated, had elevated glycemia (108 vs. 88 mg/dL), uric acid (11 vs. 6 mg/dL), and C-Reactive protein (45 vs. 7 mg/L). They also had more renal dysfunction (41% vs. 7%), electrolytes imbalance (sodium, potassium, and calcium), hepatic dysfunction (elevated ALT, bilirubin, and INR) and more hematologic disorders (WBC, red cell indices).

### 3.2. Blood Concentrations of Heavy Metals and Dilated Cardiomyopathy

Geometric means of cobalt, copper, and arsenic were significantly higher in patients than in controls, and beyond the reference values. Conversely, vanadium, zinc, antimony, and barium were significantly higher in controls than in patients, and also beyond the reference values. Higher Zn/Cu ratio was found in controls (Table 1).

DCM prediction models with adjustment for age, sex, education, and renal function (Table 2) revealed that higher blood levels of arsenic and copper were independently associated with DCM. However, a high Zn/Cu ratio, and high concentrations of vanadium, zinc, antimony, and barium, were not associated with DCM.

Probability curves of the likelihood of developing DCM as a function of heavy metals blood concentrations are shown in Figure 2. A sharp rise of DCM probability at levels of blood copper between 100 and 150 µg/dL was observed. Above 150 μg/dL, the probability was almost 100%. The same trend was seen with arsenic above 0.75 μg/dL. In contrast, DCM probability decreases as blood concentrations of vanadium (C), zinc (D), antimony (E), and the Zn/Cu ratio increase.

### 3.3. Urine Concentrations of Heavy Metals and Dilated Cardiomyopathy

As summarized in Table 3, the concentrations of beryllium, chromium, manganese, cobalt, copper, zinc, selenium, cadmium, tin, antimony, thallium, and uranium were well beyond the reference values and significantly higher in patients than in controls.

DCM prediction models with adjustment for age, sex, or educational level showed that higher levels of chromium, copper, zinc, selenium, cadmium, antimony, thallium, and uranium in urine were significantly associated with DCM (Table 4). At concentrations (in μg/g) of creatinine above 1 for chromium, 20 for copper, 600 for zinc, 30 for selenium, 2 for cadmium, 0.2 for antimony, 0.5 for thallium, and 0.05 for uranium, the probability of DCM was almost maximum (Figure 3). Significant correlations were seen between blood and urinary concentrations of several of the heavy metals of interest (Appendix A).

## 4. Discussion

This is the first study to look at estimated Glomerular Filtration Rate (eGFR)-adjusted associations between idiopathic DCM and high blood or urinary concentrations of several heavy metals in a rarely documented context of concurrent and multiple exposures. Blood and urinary concentrations for most of the metals of interest were above reference values [38,44,47]. In addition, significant correlations were found between blood and urine concentrations of metals, underlining complex interactions in the biology of heavy metals including those with known beneficial biological properties strengthening the validity of study procedures and measurements [48,49].

Blood and/or urinary concentrations of several metals were above reference values both in subjects with DCM and those presumably healthy though significantly higher in subjects with DCM when considering metal concentrations in urine. In the DRC contexts of poorly regulated mining and consistent with previous studies [14,19,38], our findings possibly reflect community-wide concurrent and multiple exposures to toxic compounds. We also showed that DCM probability almost reaches the maximum beyond specific thresholds for blood arsenic and copper concentrations or urinary concentrations of chromium, cadmium, antimony, thallium, and uranium.

The blood arsenic concentration found in our subjects (0.39 µg/dL) far exceeds what has been reported in the general population of France (0.17 µg/dL) [47], Brazil (0.11 µg/dL) [50], Pakistan (0.21 µg/dL) [51], and China (0.23 µg/dL) [52]. Whether the association between DCM probability and higher blood levels of arsenic implies the later may contribute to the etiology of DCM is not known, but remains a possibility. Experimental studies in rodents, however, have demonstrated the toxic effect of arsenic on myocardial tissue through the inhibition of anti-oxidative stress defense enzymes [53,54]. Arsenic exposure has been associated with cardiopathologic effects, including ischemia, arrhythmia, and heart failure [55]. Possible mechanisms include increased oxidative stress, depletion of antioxidant status, DNA fragmentation, apoptosis by mitochondrial disruption, caspase activation, MAPK signaling and p53, functional changes in ion channels, and dyshomeostasis of trace elements [55]. It has also been revealed that arsenic can induce all kinds of diseases, including heart diseases through epigenetic modifications associated with hypermethylation of genes coding for ion channels and diverse proteins of oxidative stress and energy production [56,57].

Exposure to copper is associated with increased cardiovascular risk [58]. High copper concentrations had been associated with heart failure incidence [36,59,60,61,62,63] and bad outcomes such as re-hospitalizations and deaths over a one-year follow-up [64]. Copper is an important trace element in humans that is incorporated in several enzymes of vital functions. However, at high concentrations due to permanent exposure or in Wilson’s disease, for example, its free fraction increases in such manner that its infiltrates the myocardial tissue, induces cellular toxicity, and promotes harmful free radical formation [65,66,67,68]. Although the blood concentration of copper reported in this study was slightly higher than the upper limit of the reference value (143.7 vs. 140 µg/dL) [69], the urine concentration far exceeds levels reported as reference value [44,70] and in a very high-exposure environment [38]. This high urinary concentration, which, moreover, was strongly correlated with blood concentration (r = 0.73235, *p* < 0.000), was precisely such an indication of a strong recent exposure. It was also an indication of a permanent exposure to this metal, which, ironically, is the great wealth of the KC. Because of copper and cobalt, the KC has been invaded by many mining companies that are the main polluters in the region.

The association between zinc deficiency and dilated cardiomyopathy is well documented [45,59,60,71,72,73]. Urinary concentrations of Zn in subjects with DCM was 4 times the reference value denoting very high excretion [47], a commonly noted phenomenon during heart failure. Aldosteronism and activation of atrial natriuretic peptides increase urinary excretion of zinc and loss in feces [74,75]. Aldosteronism also causes intracellular calcium overload, with the consequent induction of oxidative stress leading to necrosis of myocardial cells. Increasing intracellular calcium is coupled with zinc entry into cells to counteract the prooxidant effect of calcium overload. Aldosteronism causes acidification of the urine and a state of alkalosis, which are conducive to the urinary excretion of zinc. The inverse effects of acetazolamide would therefore be beneficial to curb this excretion [76]. Competition with heavy metals, especially with copper and cadmium, also explains its blood deficit and high urine excretion [77]. As zinc serves to maintain normal cell structure and function by its anti-oxidant and anti-inflammatory properties [71,78], both absolute and relative deficiency would impact negatively the proper function of the heart.

We found a strong correlation between blood copper and urinary zinc that illustrated the intimate relationship between the two metals. Indeed, the increase in blood copper leads to increased excretion of zinc and consequently its deficit. The copper-zinc intimacy is also illustrated by the Zn/Cu ratio. The fact that the highest Zn/Cu ratios were associated with a zero probability of DCM is in line with previous findings [62].

Urinary levels of chromium, cadmium, antimony, thallium, and uranium were significantly higher in DCM patients than controls. These toxic metals have all been implicated in cardiovascular disease. For example, the concept of chromium cardiomyopathy is discussed in a study in rats that were treated with potassium dichromate, and later showed changes in cardiac muscle and interstitial fibrosis. In addition, there was vacuolization, hemorrhages, and cell necrosis [79]. Accumulation of chromium in the myocardial tissue of patients with idiopathic DCM [33] has also been reported. Cadmium exposure may increase the prevalence of stroke and heart failure [80]. At lower concentrations than those of this study, cadmium was associated with idiopathic dilated cardiomyopathy [81]. These cadmium concentrations being obtained in essentially non-smoking patients argues in favor of exogenous (environmental) exposure.

Leishmaniasis treatment with antimony can cause serious arrhythmias such as torsade-de-pointe or, reportedly, an alteration of myocardial contractility through oxidative stress and disruption of intracellular calcium handling [34,35,82,83]. In our study, higher urinary concentrations of antimony were associated with DCM, supporting the above thesis on putative cardiotoxicity, and this was possibly due to yet-to-be-fully-documented environmental exposures.

Thallium, an extremely toxic metal, has the same ionic radius as potassium. Thallium follows the distribution of potassium in the cellular and extracellular compartments. The cell membrane cannot differentiate between these two ions. The Na+/K+-ATPase pump has ten times more affinity for thallium than for potassium. Thus, all potassium-dependent biological processes are altered in the presence of thallium. Through this, thallium can have immediate cardiovascular effects such as tachycardia, hypertension, ventricular fibrillation, and other electrocardiographic abnormalities. Thallium can also directly stimulate chromaffin cells and lead to a significant release of catecholamines that are deleterious for the myocardium [84,85,86,87]. Measurement of urinary thallium is a simple way to detect exposure to thallium [88]. The fact that thallium was found in greater concentration in the patients’ urine relative to controls suggests a participation of this metal in DCM morbidity.

Known studies in animal models exposed to uranium have not shown a cardiovascular effect [89,90]. However, numerous signs have been reported in the course of acute uranium poisoning, including myocarditis resulting in episodic atrial flutter [91]. Patients’ urinary uranium was approximately 2.5 times that of controls and reference values. This may suggest a harmful effect of this metal on the myocardium.

It is difficult to determine whether blood and/or urinary concentrations of all the above culprits contribute to DCM morbidity due to challenges in analytical methods for multiple exposures and because of mutual influences heavy metals exert on each other biology in concurrent exposures. A relatively small sample size and the lack of normal reference for the KC population are the main limitations of the present study. A larger sample size would have allowed studying the interactions between metals and controlling for other important clinical and/or biological features including but not limited to inflammation and/or liver functions. Nevertheless, our findings are consistent with those from previous studies that have shown ubiquitous and concurrent exposures to potentially cardiotoxic metals in the KC mining area. GFR-adjusted associations between DCM and concentrations of several of the above-mentioned heavy metals warrant public health measures to mitigate exposures to the toxic culprits. In addition, we recommend screening for heavy metals in contexts of cardiac morbidity and obvious environmental pollution from mining and/or ore processing activities. Chelation therapies and/or supplementation with elements with beneficial benefits to cellular homeostasis should also be thoroughly assessed as direct options to prevent cardiotoxicity in such contexts.

## Figures and Tables

**Figure 1 ijerph-18-04956-f001:**
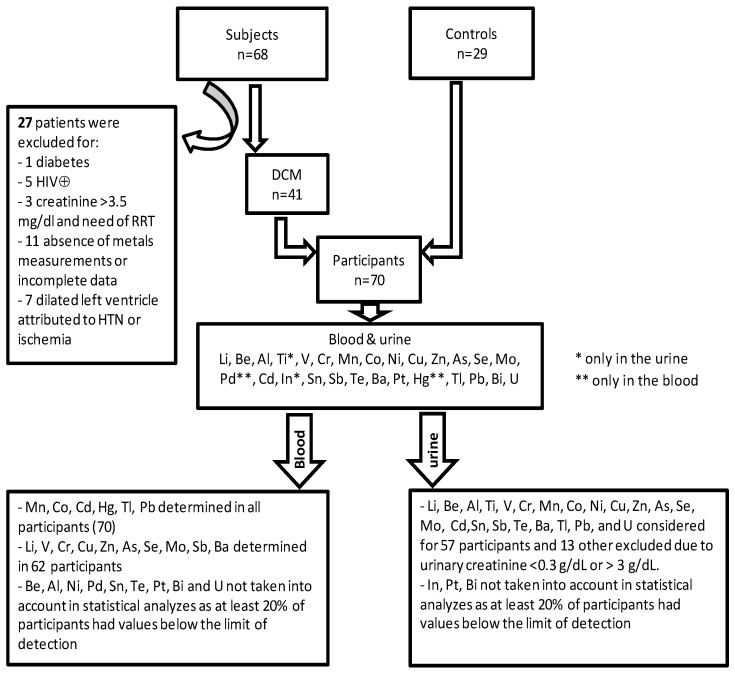
Patient selection, heavy metals measured in total blood and urine. LV denotes left ventricle, HIV⊕: human immunodeficiency virus positive test, RRT: renal replacement therapy, HTN: hypertension, DCM: dilated cardiomyopathy (idiopathic and postpartum cardiomyopathy).

**Figure 2 ijerph-18-04956-f002:**
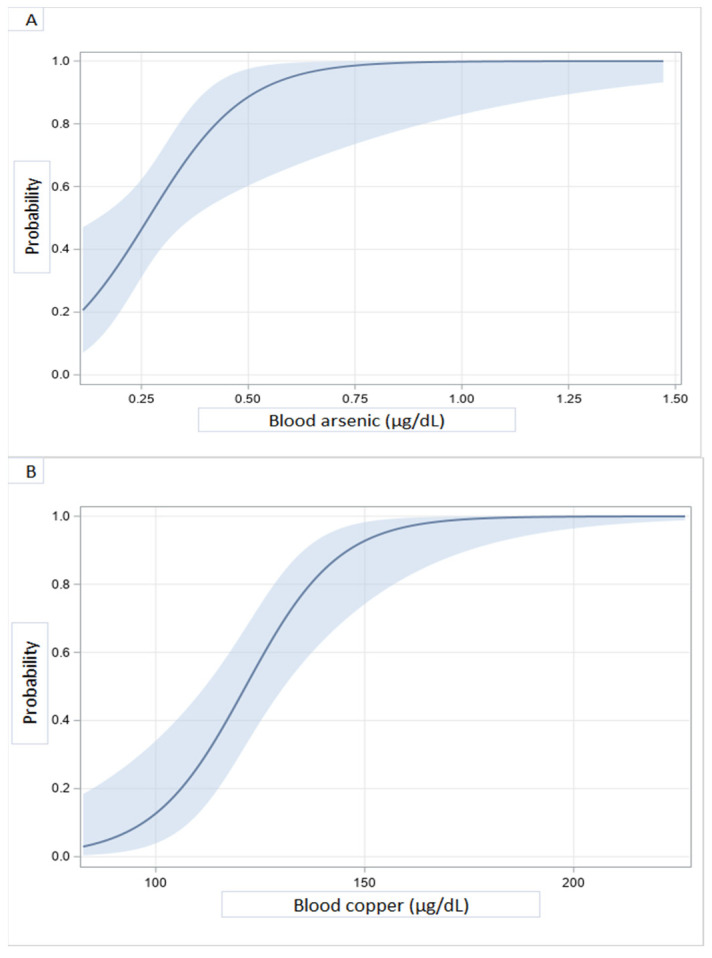
Illustrative predicted probabilities of DCM as a function of blood heavy metal concentrations: arsenic (**A**), copper (**B**), vanadium (**C**), zinc (**D**), antimony (**E**), and zinc/copper ratio (**F**).

**Figure 3 ijerph-18-04956-f003:**
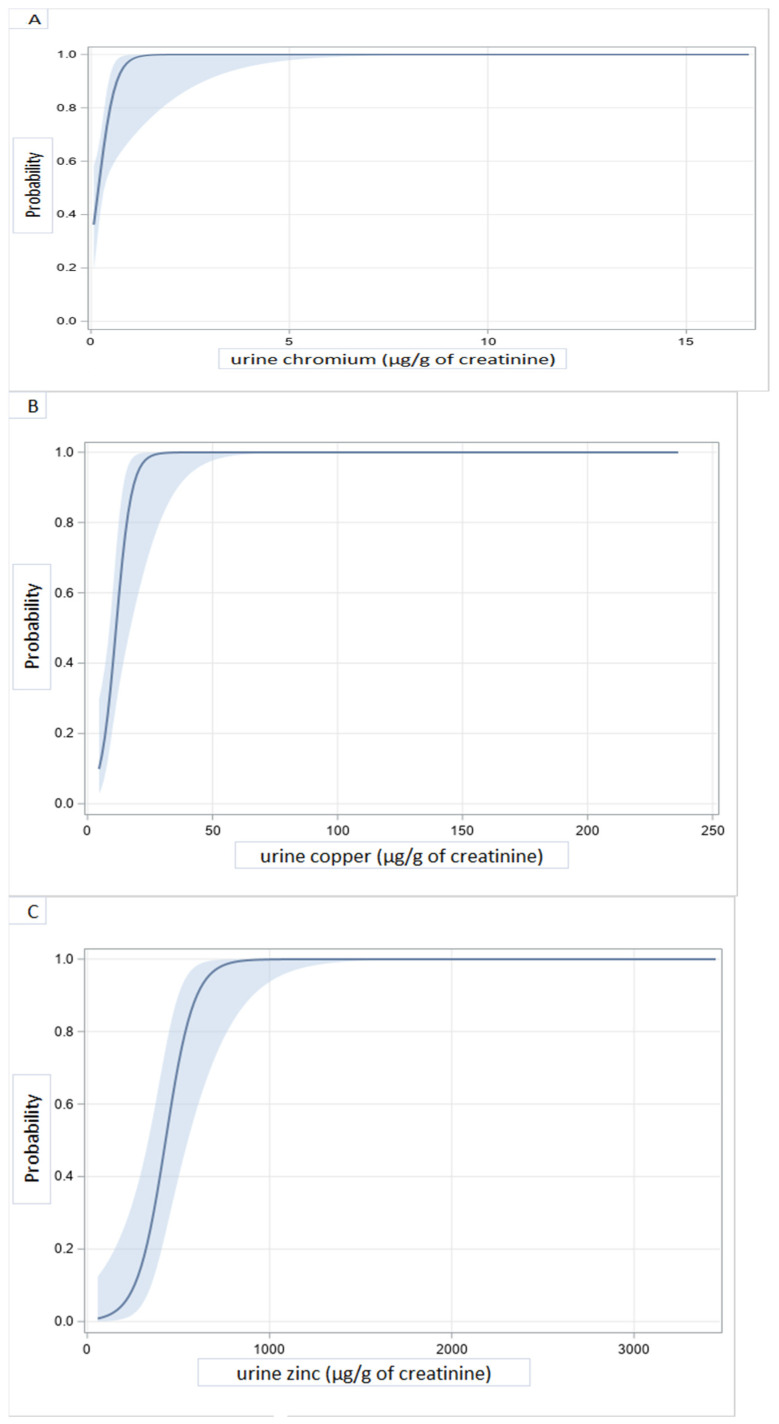
Illustrative predicted probabilities of dilated cardiomyopathy as a function of heavy metals concentrations in urine: chromium (**A**), copper (**B**), zinc (**C**), selenium (**D**), cadmium (**E**), antimony (**F**), thallium (**G**), and uranium (**H**).

**Table 1 ijerph-18-04956-t001:** Concentrations (μg/dL) of blood heavy metals in patients compared to controls.

Metal (Reference Value)	Controls (*n* = 29) GM (95% CL) Mean ± SD Median (IQR)	DCM (*n* = 41) Ø GM (95% CL) Mean ± SD Median (IQR)	GM Ratio (95% CL)	*p*
Mn 0.77 ¥	1.59 (1.41–1.81)	1.56 (1.39–1.76)	1.02 (0.86–1.22)	0.8094
	1.69 ± 0.61	1.67 ± 0.69		
	1.59 (0.58)	1.64 (0.80)		
Co 0.03 ¥	0.06 (0.04–0.08)	0.12 (0.08–0.16)	0.50 (0.31–0.82)	0.0063
	0.09 ± 0.08	0.22 ± 0.31		
	0.05 (0.07)	0.09 (0.14)		
Cd 0.04 ¥	0.16 (0.15–0.17)	0.13 (0.08–0.21)	1.24 (0.86–1.31)	0.3636
	0.17 ± 0.04	0.21 ± 0.28		
	0.16 (0.04)	0.14 (0.09)		
Hg	0.26 (0.22–0.31)	0.26 (0.22–0.31)	0.99 (0.77–1.28)	0.9663
	0.29 ± 0.15	0.31 ± 0.24		
	0.27 (0.13)	0.26 (0.13)		
Tl 0.002 ¥	0.0027 (0.0024–0.0031)	0.0028 (0.002–0.0039)	0.96 (0.67–1.38)	0.8273
	0.0029 ± 0.0012	0.0050 ± 0.0082		
	0.0027 (0.0008)	0.0036 (0.0027)		
Pb 1.88 ¥	5.65 (4.83–6.61)	6.99 (5.94–8.22)	0.81 (0.64–1.02)	0.0686
	6.13 ± 2.59	8.02 ± 4.94		
	5.83 (2.81)	6.76 (2.93)		
Li	0.21 (0.08–0.61)	0.10 (0.07–0.16)	2.08 (0.67–6.45)	0.1981
	1.11 ± 1.05	0.26 ± 0.54		
	1.22 (1.89)	0.10 (0.11)		
V 0.005 *	0.02 (0.01–0.05)	0.006 (0.004–0.01)	3.5 (1.40–8.98)	0.0086
	0.07 ± 0.07	0.02 ± 0.31		
	0.52 (0.13)	0.007 (0.008)		
Cr 0.04	0.20 (0.18–0.22)	0.18 (0.15–0.21)	1.12 (0.94–1.33)	0.2136
	0.21 ± 0.06	0.20 ± 0.13		
	0.19 (0.04)	0.17 (0.07)		
Cu 70–140 ‡	110.6 (105–116.4)	143.7 (135.1–153)	0.77 (0.71–0.83)	<0.0001
	111.51 ± 14.83	145.88 ± 25.74		
	109.71 (20.50)	146.84 (31.15)		
Zn 580.5 ¥	1196.9 (992.1–1444)	823.4 (740.3–915.8)	1.5 (1.2–1.8)	0.0009
	1331.38 ± 577.13	864.21 ± 307.60		
	1509.84 (906.91)	797.38 (230.11)		
Zn/Cu	10.8 (8.8–13.2)	5.7 (5.1–6.4)	1.9 (1.5–2.4)	<0.0001
	12.18 ± 5.44	6.08 ± 2.47		
	13.33 (8.02)	5.39 (1.95)		
As 0.17 ¥	0.23 (0.20–0.26)	0.39 (0.32–0.49)	0.58 (0.46–0.74)	<0.0001
	0.25 ± 0.09	0.47 ± 0.33		
	0.22 (0.09)	0.38 (0.23)		
Se 12.5 **	12.6 (11.8–13.4)	11.9 (11.1–12.8)	1.05 (0.9–1.2)	0.2820
	12.78 ± 2.12	12.20 ± 2.51		
	13.12 (3.34)	11.82 (3.28)		
Mo 0.1–0.3 ‡	0.18 (0.16–0.21)	0.21 (0.18–0.25)	0.87 (0.69–1.09)	0.2385
	0.20 ± 0.11	0.25 ± 0.23		
	0.17 (0.07)	0.19 (0.08)		
Sb 0.005	1.11 (0.96–1.29)	0.56 (0.49–0.64)	1.98 (1.63–2.42)	<0.0001
	1.19 ± 0.46	0.61 ± 0.29		
	1.18 (0.47)	0.57 (0.27)		
Ba 0.05–0.25	0.13 (0.09–0.16)	0.31 (0.2–0.48)	2.45 (1.50–4.01)	0.0006
	0.53 ± 0.53	0.18 ± 0.29		
	0.33 (0.68)	0.11 (0.05)		

Ø: Lithium, vanadium, chromium, copper, zinc, arsenic, selenium, molybdenum, antimony, and barium were measured in only 33 patients with DCM. ‡: WHO. Trace elements in human nutrition and health. 1996. ¥: Nisse C et al., Blood and urinary levels of metals and metalloids in the general adult population of Northern France: The IMEPOGE study, 2008-2010. Int J Hyg Environ Health. 2017;220(2 Pt B):341-63. *: Agency for Toxic Substances and Disease Registry (ATSDR). 2012. Toxicological Profile for Vanadium. Atlanta, GA: U.S. Department of Health and Human Services, Public Health Services. **: Agency for Toxic Substances and Disease Registry (ATSDR). 2003. Toxicological Profile for Selenium (Update). Atlanta, GA: U.S. Department of Health and Human Services, Public Health Service.

**Table 2 ijerph-18-04956-t002:** Adjusted models of DCM as a function of blood metal levels.

Variable	Coefficient	SE	Wald 95% Confidence Limits		Wald X^2^	*p*-Value
Arsenic Model						
Intercept	−2.84	1.65	−6.06	0.39	2.97	0.0850
As	9.04	3.44	2.28	15.79	6.88	0.0087
Age	−0.001	0.03	−0.06	0.06	0.00	0.9639
Male sex	0.48	0.70	−0.89	1.85	0.47	0.4909
Education X	−1.14	0.74	−2.59	0.29	2.43	0.1192
GFR < 60	2.21	1.00	0.24	4.17	4.83	0.0279
Copper model						
Intercept	−13.49	3.98	−21.29	−5.68	11.48	0.0007
Cu	0.09	0.02	0.04	0.14	13.07	0.0003
Age	0.02	0.03	−0.04	0.09	0.50	0.4776
Male sex	1.46	0.87	−0.24	3.15	2.85	0.0915
Education	−0.79	0.80	−2.37	0.77	0.99	0.3195
GFR < 60	1.18	1.15	−1.07	3.44	1.06	0.3042
Vanadium model						
Intercept	1.39	1.49	−1.53	4.31	0.87	0.3517
Vanadium	−23.91	8.93	−41.42	−6.40	7.16	0.0074
Age	−0.01	0.03	−0.08	0.05	0.22	0.6394
Male sex	0.30	0.68	−1.04	1.65	0.19	0.6589
Education	−1.18	0.70	−2.55	0.19	2.83	0.0925
GFR < 60	2.75	1.13	0.54	4.96	5.95	0.0147
Zinc model						
Intercept	3.36	1.86	−0.28	7.00	3.28	0.0701
Zinc	−0.003	0.0009	−0.005	−0.001	9.53	0.0020
Age	−0.02	0.03	−0.08	0.05	0.21	0.6437
Male sex	0.79	0.73	−0.63	2.22	1.18	0.2769
Education	−1.39	0.74	−2.84	0.06	3.54	0.0598
GFR < 60	3.32	1.31	0.75	5.88	6.43	0.0112
Antimony model						
Intercept	4.89	2.16	0.66	9.13	5.12	0.0236
Antimony	−4.65	1.25	−7.09	−2.20	13.87	0.0002
Age	−0.02	0.04	−0.09	0.05	0.40	0.5284
Male sex	0.58	0.82	−1.01	2.19	0.52	0.4720
Education	−1.42	0.84	−3.07	0.22	2.89	0.0892
GFR < 60	3.58	1.66	0.33	6.83	4.65	0.0310
Barium model						
Intercept	1.32	1.40	−1.43	4.06	0.89	0.3459
Barium	−2.73	1.24	−5.17	−0.29	4.81	0.0284
Age	−0.01	0.03	−0.07	0.05	0.15	0.6997
Male sex	0.19	0.65	−1.09	1.48	0.09	0.7673
Education	−1.18	0.67	−2.49	0.13	3.13	0.0769
GFR < 60	2.54	1.01	0.57	4.51	6.36	0.0117
Ratio Zinc/Copper						
Intercept	3.63	2.03	−0.35	7.61	3.20	0.0738
Zinc/copper	−0.46	0.14	−0.74	−0.19	11.01	0.0009
Age	−0.01	0.04	−0.09	0.07	0.06	0.8053
Male sex	1.44	0.89	−0.31	3.20	2.59	0.1077
Education	−1.31	0.81	−2.89	0.27	2.63	0.1050
GFR < 60	3.73	1.71	0.38	7.07	4.76	0.0291

X: Being highly educated (university or post-university) often correlates with higher socio-economic status in the KC settings.

**Table 3 ijerph-18-04956-t003:** Concentrations (μg/g creatinine) of metals in urine from DCM patients and controls.

Metal (Reference Value)	Controls (*n* = 25) GM (95% CL) Mean ± SD Median (IQR)	DCM (*n* = 32) GM (95% CL) Mean ± SD Median (IQR)	GM Ratio (95% CL)	*p*
Li 21.5	13.93 (11.98–16.19)	16.06 (12.55–20.57)	0.87 (0.65–1.15)	0.3189
	14.92 ± 6.14	20.52 ± 18.51		
	13.44 (6.04)	15.25 (13.44)		
Be	0.0009 (0007–0.0012)	0.0029 (0.0015–0.0056)	0.31 (0.16–0.62)	0.0015
	0.0011 ± 0009	0.041 ± 0.15		
	0.0008 (0004)	0.0022 (0.0059)		
Al 2.04	10.62 (7.97–14.15)	15.62 (10.22–23.88)	0.68 (0.41–1.12)	0.1292
	14.23 ± 15.39	97.42 ± 438.12		
	9.59 (6.42)	11.25 (10.62)		
Ti	30.87 (25.19–37.82)	31.13 (23.07–42.02)	0.99 (0.69–1.41)	0.9612
	34.49 ± 16.15	39.95 ± 25.09		
	29.40 (25.48)	34.43 (29.65)		
V 0.22	1.46 (0.73–2.93)	1.53 (0.91–2.57)	0.96 (0.42–2.19)	0.9178
	3.33 ± 3.49	2.95 ± 3.08		
	2.07 (3.24)	1.92 (2.67)		
Cr 0.11	0.18 (0.14–0.22)	0.36 (0.24–0.54)	0.49 (0.32–0.78)	0.0033
	0.20 ± 0.13	1.00 ± 2.91		
	0.16 (0.08)	0.29 (0.39)		
Mn <0.043	0.12 (0.06–0.21)	0.39 (0.19–0.81)	0.29 (0.11–0.78)	0.0145
	0.28 ± 0.37	3.36 ± 8.65		
	0.11 (0.27)	0.29 (1.05)		
Co 0.2	1.18 (0.73–1.92)	2.75 (1.98–3.82)	0.43 (0.25–0.75)	0.0034
	2.45 ± 3.58	4.31 ± 5.08		
	0.97 (1.84)	2.24 (3.46)		
Ni 1.79	1.19 (0.94–1.51)	1.49 (1.16–1.92)	0.79 (0.57–1.13)	0.1967
	1.39 ± 0.75	1.87 ± 1.33		
	1.18 (1.24)	1.57 (1.44)		
Cu 6.84	7.38 (6.45–8.43)	25.76 (19.93–33.29)	0.29 (0.22–0.38)	<0.0001
	7.78 ± 2.81	32.48 ± 23.22		
	7.06 (2.48)	28.52 (25.87)		
Zn 246	221.10 (176.70–276.07)	1033.80 (818.5–1305.7)	0.21 (0.15–0.29)	<0.0001
	250.38 ± 116.76	1256.08 ± 820.55		
	256.67 (167.09)	1007.14 (821.22)		
As 13.7	20.36 (14.99–27.65)	27.06 (20.41–35.89)	0.75 (0.49–1.13)	0.1689
	27.19 ± 24.59	35.21 ± 24.55		
	19.97 (20.48)	29.06 (29.68)		
Se 21.6	17.42 (15.48–19.59)	24.53 (21.70–27.73)	0.71 (0.59–0.84)	0.0001
	18.09 ± 5.07	25.98 ± 9.23		
	18.54 (5.63)	22.85 (12.10)		
Mo 29.8	60.46 (41.95–87.13)	55.17 (38.23–79.60)	1.09 (0.65–1.83)	0.4882
	84.94 ± 74.05	92.35 ± 137.29		
	65.18 ± 38.00	49.98 (54.58)		
Cd 0.22	0.61 (0.48–0.76)	1.48 (1.15–1.90)	0.41 (0.29–0.58)	<0.0001
	0.72 ± 0.54	1.94 ± 1.96		
	0.57 (0.41)	1.44 (1.29)		
Sn 0.35	0.23 (0.19–0.29)	3.59 (1.79–7.19)	0.07 (0.03–0.13)	<0.0001
	0.26 ± 0.14	11.24 ± 14.55		
	0.21 (0.11)	6.86 (16.66)		
Sb 0.04	0.05 (0.04–0.06)	0.08 (0.05–0.11)	0.64 (0.42–0.97)	0.0351
	0.05 ± 0.04	0.11 ± 0.09		
	0.046 (0.019)	0.08 (0.14)		
Te 0.14	0.23 (0.19–0.27)	0.23 (0.19–0.28)	1.01 (0.78–1.32)	0.9417
	0.25 ± 0.09	0.27 ± 0.17		
	0.21 (0.14)	0.21 (0.24)		
Ba 1.86	1.27 (0.94–1.72)	1.65 (1.04–2.57)	0.77 (0.45–1.33)	0.3442
	1.61 ± 1.19	3.54 ± 4.96		
	1.16 (0.74)	1.89 (3.98)		
Tl 0.18	0.14 (0.11–0.17)	0.23 (0.18–0.29)	0.59 (0.42–0.83)	0.0031
	0.16 ± 0.11	0.29 ± 0.19		
	0.13 (0.05)	0.25 (0.16)		
Pb 1.78	1.76 (1.38–2.26)	1.24 (0.90–1.69)	1.42 (0.95–2.15)	0.0888
	2.09 ± 1.28	1.91 ± 2.50		
	1.68 (1.28)	1.24 (1.22)		
U <0.007	0.008 (0.006–0.0105)	0.02 (0.016–0.031)	0.37 (0.24–0.55)	<0.0001
	0.11 ± 0.01	0.03 ± 0.04		
	0.007 (0.004)	0.02 (0.03)		

**Table 4 ijerph-18-04956-t004:** Adjusted models of DCM prediction by urine metal concentrations (*N* = 57).

Variable	Coefficient	SE	Wald 95% Confidence Limits		Wald X^2^	*p*-Value
Chromium model						
Intercept	−2.32	1.29	−4.85	0.21	3.24	0.0717
Chromium	4.69	2.09	0.60	8.78	5.05	0.0246
Age	0.02	0.02	−0.02	0.07	0.90	0.3435
Male sex	1.04	0.69	−0.32	2.40	2.24	0.1344
Education X	−0.86	0.65	−2.13	0.41	1.76	0.1847
Copper model						
Intercept	−5.87	2.15	−10.10	−1.65	7.44	0.0064
Copper	0.32	0.11	0.10	0.54	8.32	0.0039
Age	0.03	0.04	−0.04	0.10	0.69	0.4058
Male sex	0.88	0.95	−0.98	2.74	0.85	0.3556
Education	0.08	0.94	−1.76	1.93	0.01	0.9313
Zinc model						
Intercept	−4.88	2.65	−10.07	0.29	3.41	0.0648
Zinc	0.02	0.01	0.01	0.03	9.45	0.0021
Age	−0.05	0.06	−0.17	0.07	0.67	0.4146
Male sex	−0.81	1.26	−3.28	1.67	0.41	0.5231
Education	0.95	1.31	−1.62	3.52	0.53	0.4684
Selenium model						
Intercept	−7.39	2.41	−12.13	−2.66	9.38	0.0022
Selenium	0.24	0.08	0.09	0.39	9.84	0.0017
Age	0.05	0.03	−0.01	0.10	3.11	0.0780
Male sex	1.51	0.78	−0.02	3.05	3.74	0.0533
High education	−1.28	0.74	−2.73	0.16	3.03	0.0818
Cadmium model						
Intercept	−2.19	1.35	−4.83	0.46	2.63	0.1051
Cadmium	2.47	0.87	0.78	4.17	8.19	0.0042
Age	−0.01	0.03	−0.07	0.05	0.16	0.6850
Male sex	1.39	0.77	−0.11	2.91	3.29	0.0697
Education	−0.83	0.70	−2.20	0.55	1.38	0.2401
Antimony model						
Intercept	−2.39	1.33	−4.99	−0.22	3.23	0.0723
Antimony	14.59	6.11	2.61	26.56	5.70	0.0170
Age	0.03	0.02	−00.1	0.08	1.97	0.1602
Male sex	0.56	0.64	−0.69	1.82	0.78	0.3785
Education	−0.77	0.65	−2.04	0.51	1.39	0.2380
Thallium model						
Intercept	−3.13	1.40	−5.88	−0.38	4.96	0.0259
Thallium	8.51	3.32	2.01	15.01	6.58	0.0103
Age	0.03	0.03	−0.02	0.08	1.43	0.2324
Male sex	1.36	0.74	−0.08	2.80	3.42	0.0645
Education	−1.25	0.69	−2.59	0.09	3.33	0.0681
Uranium model						
Intercept	−3.78	1.62	−6.95	−0.61	5.48	0.0193
Uranium	85.43	35.37	16.11	154.74	5.83	0.0157
Age	0.04	0.03	−0.01	0.09	2.62	0.1055
Male sex	1.59	0.78	0.07	3.10	4.19	0.0408
Education	−0.92	0.71	−2.32	0.47	1.68	0.1944

X: Being highly educated (university or post-university) often correlates with higher socio-economic status in the KC settings.

## Data Availability

The study was conducted according to the guidelines of the Declaration of Helsinki, and was approved on 9 August 2017 by the Committee of Medical Ethics of the University of Lubumbashi (UNILU/CEM/075/2017).

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
