# Peer review of "Concurrent Heavy Metal Exposures and Idiopathic Dilated Cardiomyopathy: A Case-Control Study from the Katanga Mining Area of the Democratic Republic of Congo"

_ijerph, 2021, doi:10.3390/ijerph18094956_

Round 1

Reviewer 1 Report

Recommendation: Minor revision

In this paper, the concentrations of 27 heavy metals in the blood and/or urine of 41 patients with dilated cardiomyopathy (DCM) and 29 healthy subjects from Katanga copper strip (KC), Democratic Republic of Congo were determined. The author identified the associated heavy metal species that may influence the incidence of DCM and found that simultaneous and repeated exposure to heavy metals in excess of permissible levels was associated with an increased likelihood of DCM. It still exists some problems, although the authors have done many works. I have some comments and suggestions, and would like to see the authors’ response to the following comments and suggestions:

  1. In this paper, all subjects have statistical education level, and the education level is also used in the DCM model. Compared with other indicators, education level seems to have no obvious correlation with physical function. Please explain the significance of the following statistics of education level.
  2. In Section 2.2 of the article, ‘Palladium and mercury were only measured in the blood while titanium and indium were only measured in the urine’, whether this phenomenon which can only be detected in blood or urine is normal.
  3. In Section 3.2, ‘DCM probability decreases as blood concentrations of vanadium (2C), zinc (2D), antimony (2E) and the Zn/Cu ratio increase’ is mentioned in the article. What is the reason for this phenomenon? Could you please make a corresponding explanation?
  4. The article mentions the zinc/copper ratio and suggests that an increase in the zinc/copper ratio can reduce the incidence of DCM. Does the ratio of zinc/copper have a requirement for the copper content? In the paper, it was indicated that the increase of copper concentration would increase the incidence of DCM, and whether the presence of zinc would inhibit the effect of copper. Please give a corresponding explanation or reason.
  5. In this paper, the relationship between heavy metals and DCM is analyzed, and it is suggested that the author may add some analysis on the morphology of heavy metals.
  6. The quality of the figures in the article needs to be improved, as shown in Figure 2 and Figure 3. The text in the figures is hard to read, maybe the author can try to adjust the resolution or make the text in the figures bigger.
  7. The unit writing needs to be checked. For example, in ‘4. Discussion’ part, ‘France (0.17 µg/dL), Brazil (0.11 µg/dL), Pakistan (0.21µg/dL) and China (0.23µg/dl)’.

Reviewer 2 Report

This is a cross-sectional study that enrolled consecutive subjects who visited the Authors’ Institution and they were divided into the patients with non-ischemic cardiomyopathy and those without any cardiovascular deficit (control). Finally, 41 DCM and 29 control subjects were included in the analysis. DCM patients showed higher Co, Cu, and As levels and lower V, Zn, Zn/Cu, Sb, and Ba levels in the blood compared to the control. Urine levels of most of the heavy metals were elevated in DCM. Multivariable models demonstrated that blood or urine concentrations of some heavy metals are independent predictor of DCM.

Major comments:

Ethical consideration. The Reviewer cannot find the comments on ethics in this study. Please provide the information of informed consent, IRB, and  handling of personal information etc.  

Please specify the study design. The Reviewer assumes that the Authors retrospectively enrolled the subjects.

It looks like there is a close relationship between Cu and Zn from the physiological aspect. How about the correlation between Cu and Zn levels? Is there good inverse correlation?

Minor comments:

Line 195. “showed” may be corrected to “shown”

Reviewer 3 Report

Dear Authors,

You really did a good job.

The research focus is very relevant and highly beneficial to the health of the residents of the area where it was carried out. It has a lot of information for the readers as well and will contribute substantially to the body of knowledge. I used "track changes" to make very few corrections. You will find them in the attached document.

I will also like to request you to add more flesh to your introduction, especially as it regards your selected heavy metals of concern.

Regards.

Round 2

Reviewer 2 Report

The Authors overall responded to the Reviewer's comments, although the findings on the  association between Cu and Zn can be described in the paper.